# Translation and interpreting teachers' perceptions of dilemma and needs in their professional development

Yi Liu, Jianyu Liu 🔵 *

College of Foreign Studies, Liaoning University, Shenyang, Liaoning Province, P. R. China

* ljy_0923@163.com

**Data Availability Statement:** All relevant data are within the paper and its Supporting Information files.

## Abstract

Since Chinese universities launched the postgraduate program of the Master of Translation and Interpreting (MTI) in 2007, approximately 300 high education institutions in China have been authorized to offer the MTI program. Behind the drastic springing-up of MTI programs during the decade, MTI teachers' professional development draws the attention of MTI administrators and researchers. This study adopted a mixed-method of a large-scale survey among 514 MTI teachers across China and a qualitative interview study of seven participants and discussed MTI teachers' perceptions of dilemmas and inner-world needs in their professional development. The findings indicate that MTI teachers' dilemmas arise from the contradiction and entanglement in three mutually repulsive sectors of their professional development, i.e., teaching, research, and translation/interpreting practice, which hinder their professional development. And on the basis of the analysis, the present study proposes a synchronized "three-in-one" mechanism model with encouraging policies and environments as lubricant for the sustainable development of MTI teachers in the future, and it is hoped that this empirical research would provide some practice implications for the professional development of translation and interpreting teachers in China and beyond.

## Introduction

With the increasing development of globalization and intercultural communication, there is high demand in China for skilled professional translators and interpreters for the purpose of introducing Chinese sociocultural and economic development to the world [1]. In such a context, the Academic Degree Committee under the State Council of the People's Republic of China launched a new postgraduate program in 2007, i.e. Master of Translation and Interpreting (MTI) to meet the demand. Ever since the MTI program grows vigorously among Chinese universities, by the end of 2023, 316 high education institutions in China have been authorized to establish the MTI program, and the number is still rising. Behind the drastic springing-up of MTI during the decade, some potential problems emerge and draw the attention of MTI policy-makers, teachers, and researchers. At the first "International Forum on the Education and Development of Translation Talents" in 2015, Zhou Mingwei, director of the China

**Funding:** Yi Liu received funding from Postgraduate Education Reform Project of Liaoning Province 2022 (Grant Number: LNYJG2022009). The funders had no role in study design, data collection and analysis, decision to publish, or preparation of the manuscript. URL: http://info.neu.edu.cn/_upload/article/files/c2/a7/ec6f61ee4b2da43f551d89398d29/7c84f1dc-adc7-43d6-b9e5-a8ef4386df6b.pdf).

**Competing interests:** The authors have declared that no competing interests exist.

Foreign Languages Publishing Administration, pointed out that the translation and interpreting (T&I) education does not match the speed of Chinese economic and social development, and the key point to this disparity is the shortage of highly qualified T&I teachers/trainers. According to "The Notice on Trial Running of MTI Programs" issued by the Academic Degree Committee under the State Council of China, the benchmark for a qualified MTI teacher is the completion of formal translation practice of at least 300,000 words or at least 20-hour real interpreting tasks. However, for a newly established professional post-graduate program, most MTI teachers were shifted from the academic postgraduate programs of linguistics, applied linguistics, or literature and did not have sufficient translation or interpreting practice and professional expertise essential to the practice-oriented post-graduate program [2, 3]. This insufficiency brings criticism on the professional development of MTI teachers, which are unanimously perceived as the primary guarantee and core driving force for sustainable development of MTI education. Understanding teachers' anxieties and inner-world needs is the key to promote their sustainable professional development [4, 5]. This study, therefore, sets out to explore MTI teachers' perceptions of dilemmas and needs in their professional development. It is hoped that the present descriptive study would provide empirical evidence for further exploration of T&I teachers' professional development.

## Literature review

Recent developments in the field of professional T&I education aroused research interest in T&I teachers. A number of studies examined T&I teachers' general professional status quo, such as knowledge structure [6], competences [7–9], and quality [10–12]. More recent attention has focused on cognitive and psychological issues, exploring T&I teachers' agency [13, 14], beliefs and practices [15–19], as well as roles and identities [20–22]. There are also publications on T&I teacher education and professional development mode. Tao [23] pointed out that T&I teachers are expected to be good at research, teaching, and T&I practice. Kang & Shi [24] conducted a retrospective cohort study on unbalanced factors among interpreting teaching, practice, and research, and put forward a "Practice-Teaching-Research" model for interpreting teachers' professional development. However, previously published works on T&I teachers' professional development are mostly retrospective, theoretical, and speculative top-down studies that prescribe what educational and professional qualifications that T&I teachers should have, but there are few bottom-up empirical studies on the personal aspects of them [7], especially on their anxieties and inner-world needs.

As previously mentioned, compared to the teachers of other well-developed research-oriented postgraduate programs, the "green-handed" MTI teachers, mostly shifted from the teaching of other academic programs, tend to experience more dilemmas and challenges in their professional development. As faculty members of Chinese universities, MTI teachers are expected to have not only high academic research engagement [25], but also high engagement in professional practice of translation and interpreting [19]. The disparity between these two expectations, research engagement and practice engagement, impeded teachers' academic promotion and development and induced more anxiety and dilemmas. Hence, T&I teachers' professional development mode and dilemmas and needs in their professional development become a key research issue in the T&I education field. In recent years the author and her colleagues attempt to address the issue to explore Chinese MTI teachers' dilemmas and needs in their professional development. For example, Liu & Xu [3] comprehensively described the current situation of MTI teachers in age, professional titles and academic qualifications, and the barriers in their teaching, research and T&I practice. In terms of professional development needs, Liu & Zhang [26] conducted a qualitative survey with 103 MTI teachers about the

factors influencing their development and development needs. It found that teaching, T&I practice and research constitute "three-in-one" pattern of MTI teachers' professional development as well as their competence framework. This study provides valuable insights for MTI teachers' professional development. However, a qualitative method is unsuitable for obtaining a holistic picture of difficulties and needs in the professional development of a larger group of MTI teachers. The study we present here, seeks to examine the perceptions of dilemmas and needs of MTI teachers induced by challenges and difficulties in their professional development and try to construct a facilitative system to empower MTI teachers' sustainable professional development. We approach this task on a larger scale than previous research and attempt to address the following two questions:

1. What are MTI teachers' perceptions of the challenges and dilemmas in teaching, research, and T&I practice?

2. What are MTI teachers' perceptions of the needs in their professional development?

## Methodology

### Quantitative research instrument

As mentioned above, since by the time the project was conducted, there was no sufficient relevant empirical research on MTI teachers for reference. Hence, an open-ended questionnaire survey was first adopted to depict a general picture of the MTI teachers' current situation. An exploratory prior stage for a study is suitable for the cases when the understanding of the certain issue is not very clear, when in-depth discovery and exploration of some special issues is required, and when the opinions and beliefs of some particular groups are to be inquired into [27]. This pilot qualitative exploratory study was conducted and published by the first author and her colleague (see Liu and Zhang [26] for detail).

On the basis of the pilot qualitative study, an anonymous self-completion questionnaire was designed for a large-scale survey. It covered participants' understanding of informed consent of the research so that the participants first clarified the purpose of this study and ensured that the information they provided would be strictly confidential and used only for this study. The final version of the quantitative questionnaire in the present study includes two parts, i.e., demographic characteristics of the research participants and relevant dimensions regarding MTI teachers' professional development. The main dimensions involved in this report include the degree of MTI teachers' perceptions of the challenges and dilemmas in teaching (13 items), perceptions of factors affecting their T&I practice (9 items) as well as academic research engagement (9 items), perceptions of the needs of their professional development (9 items), perceptions of the needs of in-service training content (9 items) and modes (8 items). The 57-item adapted survey was scored using a 5-point Likert scale ranging from 1 (strongly disagree) to 5 (strongly agree) and was validated in the target sample, then utilized to examine MTI teachers' perception of dilemma and needs in their professional development. After preliminary predictions and multiple revisions, the Cronbach's Alpha of the finalized questionnaire reached 0.939, and the six dimensions ranged from 0.808 to 0.926, indicating acceptable reliability and good structure validity.

### Quantitative data collection and analysis

The quantitative survey adopted the method of stratified sampling in data collection. The participants consist of 514 MTI teachers from 32 provinces, autonomous regions, and

municipalities in China, who were recruited between December 28, 2018 and January 10, 2020. The ethical committee of College of Foreign Studies of Liaoning University approved the project.

The regional distribution of the participants is basically in line with the proportion of the number of MTI universities of the corresponding region to the total number of MTI universities. Besides, the types of universities, gender, age, academic title, and educational background were also considered in sampling, and the distribution fits the status quo of MTI teachers in China (see Table 1). Hence, the participants have a high degree of representation of the population of MTI teachers.

The quantitative survey was conducted via an online questionnaire service system called Wenjuanxing (www.wjx.com), which lasted for five months. The data generated by the questionnaire were numerical, and descriptive statistics were performed through SPSS 26.

## Qualitative interview as triangulation

In addition to the quantitative survey, this study also adopted qualitative interviews as supplementary to triangulate the findings. The mixed approach has been increasingly used and

**Table 1. Demographic information of participants.**

| Demographics | | N | % |
|---|---|---|---|
| Gender | Male | 219 | 42.61 |
| | Female | 295 | 57.39 |
| Years of MTI teaching | 0–3 years | 234 | 45.53 |
| | 4–6 years | 59 | 11.48 |
| | 7–9 years | 141 | 27.43 |
| | >10 years | 80 | 15.56 |
| Academic title | Teaching Assistant | 16 | 3.11 |
| | Lecturer | 117 | 22.76 |
| | Associate Professor | 237 | 46.11 |
| | Professor | 126 | 24.51 |
| | Other cases | 18 | 3.50 |
| Educational background | Bachelor | 38 | 7.39 |
| | M.A. | 273 | 53.11 |
| | Ph.D. | 203 | 39.50 |
| Types of universities | Comprehensive univ. | 196 | 38.13 |
| | Foreign language univ. | 53 | 10.31 |
| | Normal univ. | 79 | 15.37 |
| | Univ. of natural sciences | 135 | 26.26 |
| | Univ. of social sciences/liberal arts | 21 | 4.09 |
| | Univ. of economics, political science, and law | 13 | 2.53 |
| | Univ. of arts and physics | 2 | 0.39 |
| | Other cases | 15 | 2.92 |
| Regional distribution | North China | 110 | 21.40 |
| | Northeast China | 127 | 24.71 |
| | East China | 119 | 23.15 |
| | South China | 34 | 6.62 |
| | Central China | 51 | 9.92 |
| | Southwest China | 35 | 6.81 |
| | Northwest China | 38 | 7.39 |

**Table 2. Basic information of the seven interviewed teachers.**

| | Gender | Professional title | Types of Universities | Age | Degree | Major | T&I practice experience |
|---|---|---|---|---|---|---|---|
| T1 | Male | Professor | Normal univ. | 50s | Master | FLL | Yes |
| T2 | Male | Lecturer | Comprehensive univ. | 40s | Master | FLL | No |
| T3 | Female | Associate Professor | Comprehensive univ. | 40s | Doctor | Translation | No |
| T4 | Male | Associate Professor | Comprehensive univ. | 40s | Doctor | Translation | No |
| T5 | Male | Professor | Foreign language univ. | 50s | Master | Translation | Yes |
| T6 | Female | Lecturer | Normal univ. | 30s | Master | Translation | No |
| T7 | Female | Lecturer | Univ. of natural sciences | 30s | Master | FLL | No |

Note. FLL = Foreign Languages and Literatures

accepted to conduct social research due to its advantage in understanding the phenomena of the social world, seeing this world through complex lenses, and using eclectic methodologies that better respond to the multiple stakeholders of policy issues [28] (p. 455). At the end of the quantitative questionnaire, we also collected participants interested in the follow-up qualitative interview. If they were willing to participate in our follow-up interview, they would leave their contact information (e.g., telephone number, WeChat, or email) in the questionnaire. Mindful of gender, professional title, age, geographical distribution, and types of universities, 7 participants of the 514 MTI teachers were eventually selected for qualitative interview (see Table 2). Semi-structured interviews with those MTI teachers were conducted in Mandarin for 1.5 hours on average (face-to-face, over the telephone, or via WeChat, a social media platform) and audio-recorded with the participant's permission. The interview questions were designed in reference to the six dimensions mentioned above of MTI teachers' perceived difficulties and needs in their professional development. The qualitative interview data were transcribed in textual form and then coded through NVivo 12 after iterative reading and reflection by the researchers.

## Findings and discussion

In this section, we will report the findings and discussion on the basis of the quantitative survey, and with the data collected in both the previous qualitative questionnaire and afterward interview as supplementary evidence. As Kelly [7, 29] suggested, a competent T&I trainer needs to have three different areas of competence and expertise, i.e., professional T&I practice, T&I studies as an academic discipline, and teaching skills. MTI teachers in Chinese universities experienced various difficulties and challenges in developing these types of competency that they are expected to have, which would be considered as major causes of the dilemmas in their professional development.

## MTI teachers' perceptions of dilemmas in teaching practice

Teaching is the essential and obligatory responsibility of all teachers. The questionnaire data shows that about 28% of teachers are satisfied with the current situation of MTI teaching in their universities, but nearly 72% of teachers are neutral or unsatisfactory (see Table 3 for details). The analysis of the questionnaire indicates the challenges that MTI teachers experienced in their teaching practice.

As shown in Table 4, the primary difficulty of the MTI teachers in teaching practice is the challenges caused by "varying levels of MTI students' language proficiency" (Q4-3). In the qualitative questionnaire, most MTI teachers state that the quality of the students enrolled in

**Table 3. MTI teachers' satisfaction degree with the current status of MTI education.**

| Degree of Satisfaction | N | % |
|---|---|---|
| Very dissatisfied | 34 | 6.61 |
| Dissatisfied | 81 | 15.76 |
| Neutral | 255 | 49.61 |
| Satisfied | 131 | 25.49 |
| Very satisfied | 13 | 2.53 |
| Total | 514 | 100.00 |

the program is one of their concerns, especially students' professional competence and competitiveness. There is still an apparent disparity between the prescriptive objectives of the program and market demand. Some teachers expressed their concerns about the unsatisfactory qualifications of MTI students, such as "unsolid language foundation" (9-T33), "lack of domain knowledge" (9-T31), and "unable to meet the need of the translation market" (7-T69). The items ranking second and third place are "insufficient time and energy" (Q4-4) and "lack of experience in T&I practice" (Q4-1). The triple obligations of teaching, research, and T&I practice which MTI teachers need to undertake simultaneously restrict each other in time and energy and impede teachers' motivation in their professional development. As a participant stated:

> On the one hand, translation practice conflicts with classroom teaching in time, and on the other hand, we are also expected to conduct research, and we do not have enough time and energy to complete all the three tasks. (T4)

In addition, as a newly-established program, most MTI teachers just switched from teaching research-oriented postgraduate courses to practice-oriented courses, and the lack of professional T&I practice experience and motivation, according to the MTI teachers, has seriously impacted the quality of their teaching. For example, one teacher said that she teaches the MTI course simply because "our supervisors asked me to take the interpreting course since no one would like to take it" (T2), and another stated that:

**Table 4. MTI teachers' recognition of teaching problems.**

| Items | Strongly Disagree | Disagree | Neutral | Agree | Strongly Agree |
|---|---|---|---|---|---|
| Q4-1 Lack of experience in T&I practice | 14 (2.72%) | 41 (7.98%) | 117 (22.76%) | 224 (43.58%) | 118 (22.96%) |
| Q4-2 Fewer knowledge of translation market | 19 (3.70%) | 41 (7.98%) | 113 (21.98%) | 233 (45.33%) | 108 (21.01%) |
| Q4-3 Varying levels of MTI students' language proficiency | 10 (1.95%) | 8 (1.55%) | 83 (16.15%) | 247 (48.05%) | 166 (32.30%) |
| Q4-4 Insufficient time and energy | 8 (1.56%) | 22 (4.28%) | 105 (20.43%) | 248 (48.24%) | 131 (25.49%) |
| Q4-5 Unreasonable program curriculum | 10 (1.95%) | 85 (16.54%) | 209 (40.66%) | 159 (30.93%) | 51 (9.92%) |
| Q4-6 Lack of teaching resources | 13 (2.53%) | 48 (9.34%) | 145 (28.21%) | 227 (44.16%) | 81 (15.76%) |
| Q4-7 Inadequate teaching facilities | 17 (3.31%) | 98 (19.07%) | 154 (29.96%) | 178 (34.63%) | 67 (13.03%) |
| Q4-8 Weaker motivation of teaching | 17 (3.31%) | 77 (14.98%) | 163 (31.71%) | 186 (36.19%) | 71 (13.81%) |
| Q4-9 Fewer T&I practice opportunities of students | 10 (1.94%) | 54 (10.51%) | 126 (24.51%) | 208 (40.47%) | 116 (22.57%) |
| Q4-10 Conflicts between students' T&I practice and learning | 16 (3.11%) | 90 (17.51%) | 172 (33.46%) | 169 (32.88%) | 67 (13.04%) |
| Q4-11 Insufficient learning motivation of MTI students | 16 (3.11%) | 60 (11.67%) | 138 (26.85%) | 210 (40.86%) | 90 (17.51%) |
| Q4-12 Unclear subject positioning | 12 (2.33%) | 94 (18.29%) | 159 (30.93%) | 173 (33.66%) | 76 (14.79%) |
| Q4-13 Lack of greater team support | 11 (2.14%) | 43 (8.37%) | 150 (29.18%) | 201 (39.10%) | 109 (21.21%) |

*Those (teachers) who meet the academic requirements and evaluation criteria may not have much experience in T&I practice, understanding of the translation market, or knowledge about translation technology (T7).*

In addition, the statistics show that the program curriculum (Q4-5), facilities (Q4-7), and positioning (Q4-12) have not fully aligned with the emphasis of practicality and application in MTI education. Moreover, the data also indicates that MTI teachers have pressing needs for more teaching resources (Q4-6), greater team support (Q4-13), and stronger motivation of teaching (Q4-8).

## MTI teachers' perceptions of dilemmas in academic research

As university faculty members, MTI teachers are expected to be engaged in academic research [7], and research productivity is one of the major criteria for their career progression [25]. Table 5 presents MTI teachers' recognition of factors which affect their engagement in T&I research.

Of the nine factors, "difficulties of having papers published" (Q5-6) is currently the biggest obstacle for MTI teachers' research engagement. The teachers believe that translation or interpretation is an applied skill, and relevant research on translation or interpretation is "non-academic". For example, one teacher argued that "it is not easy to publish research papers on practice-oriented translation and interpretation" (T5). Moreover, the teachers complained that the acceptance rate of papers or research projects on translation and interpretation is quite low by academic journals or fund organizations, as one teacher stated, "T&I practice takes much time, and the papers written on translation or interpretation are not that theoretical and do not meet the taste of academic journals" (T4).

Secondly, "heavy teaching load" (Q5-1) is another major factor that hinders MTI teachers' research engagement. Although most MTI teachers agreed on the importance of research, they argued that "teaching load is so heavy and stressful that they do not have extra spare time and energy for research" (11-T22). In the questionnaire, MTI teachers regarded "insufficient team support" (Q5-8) and "disconnection between research and MTI teaching" (Q5-9) as additional major factors that prevented them from more research engagement. Furthermore, those who recently switched from teaching research-oriented courses such as linguistics, language education, or literature reported that "the field of their academic research is not related to the actual MTI teaching" (11-T59).

Among all the factors, the lowest mean score is for "not interested in research" (Q5-7), indicating that MTI teachers are aware of the importance of research engagement for their professional development, but some environmental factors, such as difficulty in publication,

**Table 5. MTI teachers' recognition of engagement in academic research.**

| Items | Strongly Disagree | Disagree | Neutral | Agree | Strongly Agree |
|---|---|---|---|---|---|
| Q5-1 Heavy teaching load | 7 (1.36%) | 15 (2.92%) | 73 (14.20%) | 236 (45.92%) | 183 (35.60%) |
| Q5-2 Heavy load of T&I practice | 15 (2.92%) | 96 (18.68%) | 218 (42.41%) | 133 (25.87%) | 52 (10.12%) |
| Q5-3 Lack of theories in translation studies | 25 (4.86%) | 109 (21.21%) | 190 (36.97%) | 150 (29.18%) | 40 (7.78%) |
| Q5-4 Unfamiliar with relevant research methods | 24 (4.67%) | 103 (20.04%) | 181 (35.21%) | 164 (31.91%) | 42 (8.17%) |
| Q5-5 Lack of academic resources | 23 (4.47%) | 100 (19.46%) | 156 (30.35%) | 167 (32.49%) | 68 (13.23%) |
| Q5-6 Difficulties of having papers published | 8 (1.56%) | 13 (2.53%) | 55 (10.70%) | 182 (35.41%) | 256 (49.80%) |
| Q5-7 No interest in academic research | 57 (11.09%) | 144 (28.02%) | 188 (36.58%) | 91 (17.70%) | 34 (6.61%) |
| Q5-8 Insufficient team support | 7 (1.36%) | 53 (10.31%) | 153 (29.77%) | 199 (38.72%) | 102 (19.84%) |
| Q5-9 Disconnection between research and MTI teaching | 17 (3.31%) | 59 (11.48%) | 141 (27.43%) | 192 (37.35%) | 105 (20.43%) |

complicated process of application, and heavy load of teaching demotivate their involvement in academic research. For example, T5 complained about the situation with some passiveness:

*The application process is too complicated and following the requirement of other people instead of our own interests is also very boring. . . I prefer to summarize some experiences based on my own T&I practice. I don't need to follow the academic research requirement, go through complicated application processes, and don't need to be funded financially. (T5)*

And they also express their expectation of more environmental support, such as greater teamwork support and higher resource accessibility to overcome these obstacles, which implies how important the environment is in shaping teachers' professional development [30, 31].

## MTI teachers' perceptions of dilemmas in T&I practice

The goal of MTI education is to serve the T&I industry [32]. Hence, professional translation practice is prescribed as a prerequisite for a qualified MTI teacher, the main reference for the evaluation of MTI teachers' qualifications, and the criterion of their professional promotion as university teachers [6, 7]. As Kiraly [33] argues that without professional T&I expertise or a professional self-concept, a teacher will not be able to help their students develop one. Therefore, engaging in professional practice in the mode of in-house practice or short-term practice in T&I firms or a freelance translator is perceived as an integral part of MTI teachers' professional development.

Table 6 indicates that the factor of the highest mean value affecting MTI teachers' participation in professional practice is "heavy load of research work" (Q6-2) and "heavy load of teaching tasks" (Q6-1). For example, one of the participants said:

*On the one hand, engagement in T&I practice may conflict with teaching tasks in time, and on the other hand, the university requires full-time teachers' research output. So we do not have enough energy and time for both of them. In the past few years, apart from teaching, I have been torn apart between research and T&I practice (T6).*

When speaking of their engagement in translation practice, the teachers state that lack of time and energy deprives them of the opportunities of either working in translation firms for in-house T&I practice or any other type of translation practice. In addition, the low service fee in the market further demotivates their participation in T&I practice (Q6-4). The factors of the lowest mean score were "no interest in T&I practice" (Q6-9) and "no benefits to professional

**Table 6. MTI teachers' recognition of T&I practice.**

| Items | Strongly Disagree | Disagree | Neutral | Agree | Strongly Agree |
|---|---|---|---|---|---|
| Q6-1 Heavy load of teaching tasks | 12 (2.33%) | 11 (2.14%) | 90 (17.51%) | 256 (49.81%) | 145 28.21%) |
| Q6-2 Heavy load of research work | 13 (2.53%) | 12 (2.33%) | 78 (15.18%) | 224 (43.58%) | 187 (36.38%) |
| Q6-3 Less contact with translation market | 16 (3.11%) | 23 (4.47%) | 122 (23.74%) | 243 (47.28%) | 110 (21.40%) |
| Q6-4 Low service fee in the market | 14 (2.72%) | 21 (4.09%) | 112 (21.79%) | 192 (37.35%) | 175 (34.05%) |
| Q6-5 Lack of domain knowledge | 19 (3.70%) | 48 (9.34%) | 152 (29.57%) | 234 (45.53%) | 61 (11.87%) |
| Q6-6 Lack of experience in T&I practice | 21 (4.09%) | 70 (13.62%) | 148 (28.79%) | 206 (40.08%) | 69 (13.42%) |
| Q6-7 Lack of T&I practice competence | 26 (5.06%) | 95 (18.48%) | 152 (29.57%) | 179 (34.82%) | 62 (12.06%) |
| Q6-8 No benefits to professional development | 59 (11.48%) | 135 (26.26%) | 166 (32.30%) | 113 (21.98%) | 41 (7.98%) |
| Q6-9 No interest in T&I practice | 78 (15.18%) | 170 (33.07%) | 182 (35.41%) | 62 (12.06%) | 22 (4.28%) |

development" (Q6-8). In the qualitative questionnaire, the teachers said, "we cannot learn translation only from books" (22-T80), and "high competence in T&I practice makes MTI teachers more qualified" (22-T39). It implies that MTI teachers hold the consensus that T&I practice is an indispensable part of their professional life, and teachers' experience in professional translation practice is an essential guarantee for the success of MTI education. MTI teachers have the motive to undertake more T&I practice, but under the pressure of teaching and research, as well as other environmental factors, they have to reduce the investment of time and energy into the practice.

## Perceptions of the needs of MTI teachers in their professional development

In addition to the exploration of MTI teachers' perceptions of dilemmas arising from various challenges and difficulties in teaching, research, and T&I practice, in the questionnaire, we also explore an overall picture of the needs and expectations of MTI teachers in their professional development.

Table 7 indicates that the major needs of MTI teachers are "to understand the development of the T&I industry" (Q7-8), "to have more opportunities for in-service professional training" (Q7-9), "to learn modern T&I technology" (Q7-7) and "to enhance T&I competence" (Q7-1). It implies that MTI teachers expect to learn more about the T&I industry, to expand the relevant domain knowledge, and to enhance T&I competence. The participants in the open-ended questionnaire expressed their motivation to "establish more extensive contacts with publishers and T&I firms to get real T&I jobs" (13-T66) and "to go to T&I firms for further training and experience actual T&I projects in the market" (20-T37). And in the afterward qualitative interview, MTI teachers also reiterated that they "need professional T&I practice most because the objectives of the MTI program is to cultivate talented translators or interpreters, and MTI teachers themselves should be qualified first" (T6). In addition, in the era when information technology has revolutionized many traditional industries, T&I technology likewise became a focus of attention for MTI teachers. The teachers indicated that "A more urgent need is for the training of T&I technology" (20-T81). One teacher in the interview emphasized that:

> T&I technology development is something we have to face. Though some teachers prohibited students from using machine translation in their practice, the students would use it anyhow. We should talk about how to use machine translation scientifically, how to distinguish good translation from poor translation, learn how to edit after translation, and learn how to control the quality of translation with tools and processes. We should not deny T&I technology, but be open and tolerant and keep up with the times (T2).

**Table 7. MTI teachers' professional development needs.**

| Items | Strongly Disagree | Disagree | Neutral | Agree | Strongly Agree |
|---|---|---|---|---|---|
| Q7-1 To enhance T&I competence | 8 (1.56%) | 13 (2.53%) | 68 (13.23%) | 267 (51.95%) | 158 (30.74%) |
| Q7-2 To improve academic research competence | 7 (1.36%) | 12 (2.33%) | 78 (15.18%) | 282 (54.86%) | 135 (26.26%) |
| Q7-3 To improve teaching skills | 4 (0.78%) | 14 (2.72%) | 64 (12.45%) | 292 (56.81%) | 140 (27.24%) |
| Q7-4 To achieve academic promotion | 11 (2.14%) | 31 (6.03%) | 75 (14.59%) | 226 (43.97%) | 171 (33.27%) |
| Q7-5 To enhance knowledge of translation theory | 8 (1.56%) | 23 (4.47%) | 91 (17.70%) | 278 (54.09%) | 114 (22.18%) |
| Q7-6 To enhance the education level | 16 (3.11%) | 63 (12.26%) | 122 (23.74%) | 208 (40.47%) | 105 (20.43%) |
| Q7-7 To learn modern T&I technology | 5 (0.97%) | 10 (1.95%) | 65 (12.65%) | 289 (56.23%) | 145 (28.21%) |
| Q7-8 To understand the development of the T&I industry | 5 (0.97%) | 7 (1.36%) | 60 (11.67%) | 289 (56.23%) | 153 (29.77%) |
| Q7-9 To have more opportunities for in-service professional training | 5 (0.97%) | 9 (1.75%) | 70 (13.62%) | 263 (51.17%) | 167 (32.49%) |

They believed that T&I technology is an unstoppable tidal current in the translation industry, and the comprehension and mastery of it is an essential indicator for the professionalization of MTI teachers, which is critical for the success of the market-orientated and practice-orientated MTI program [34].

In addition to the need of increasing their professional and practical T&I competence, MTI teachers also scored higher on the needs of "to improve teaching skills" (Q7-3) and "to improve academic research competence" (Q7-2), indicating that, though slightly weaker than T&I practice, the needs for increasing teaching and research ability are likewise essential for MTI teachers' professional development. When speaking of the abilities of a competent MTI teacher, the teachers in the qualitative questionnaire define that a competent MTI teacher should have "high competence in professional T&I practice, teaching and academic research" (22-T76). The item of the lowest mean among this category is "to enhance their education level" (Q7-6), indicating that the demand of MTI teachers to further their education is relatively low, and the most pressing currently for MTI teachers is to enhance the T&I practical competence rather than obtaining higher education degrees. It implies that the education level of MTI teachers in recent years has increased to a relatively satisfactory level in comparison to the situation ten years ago [3].

## Perceptions of needs of MTI teachers for in-service training programs

Among various approaches to professional development, in-service training is perceived as one of the most efficient to enhance the qualifications of MTI teachers and to ensure the success of MTI education [16, 35–37].

The statistics indicate that nearly 65% of the participants have participated in relevant in-service training. However, when inquired whether they were satisfied with the training, nearly half of them were average or unsatisfactory with the training program (see Table 8), and it implied that their expectations and needs for in-service training have not been fully accommodated. For example, a teacher mentioned his needs for long-term training programs:

*Short-term in-service training is generally ineffective and can only give teachers a skin-deep, simple conceptual understanding, which is generally not profound enough for teaching and supervising students (T6).*

In terms of training contents, Table 9 indicates that the items of the highest mean values in this section are "T&I industry information" (Q9-6), "T&I technology" (Q9-7), "strategies in T&I practice" (Q9-3), and "T&I project management" (Q9-8). Most teachers articulate that they need to learn about the T&I industry, marketing, business operation, and management to

**Table 8. MTI teachers' satisfaction degree with in-service training programs.**

| Degree of Satisfaction | N | % |
|---|---|---|
| Very dissatisfied | 10 | 1.95 |
| Dissatisfied | 17 | 3.31 |
| Neutral | 163 | 31.71 |
| Satisfied | 157 | 30.54 |
| Very satisfied | 17 | 3.31 |
| Unknown | 150 | 29.18 |
| Total | 514 | 100.00 |

**Table 9. MTI teachers' needs of training content.**

| Items | Strongly Disagree | Disagree | Neutral | Agree | Strongly Agree |
|---|---|---|---|---|---|
| Q9-1 T&I related pedagogical theories | 12 (2.33%) | 19 (3.70%) | 118 (22.96%) | 253 (49.22%) | 112 (21.79%) |
| Q9-2 T&I related pedagogical methods | 9 (1.75%) | 13 (2.53%) | 65 (12.65%) | 294 (57.20%) | 133 (25.88%) |
| Q9-3 Strategies in T&I practice | 12 (2.33%) | 16 (3.11%) | 69 (13.42%) | 256 (49.81%) | 161 (31.32%) |
| Q9-4 Theoretical knowledge of T&I studies | 12 (2.33%) | 23 (4.47%) | 111 (21.60%) | 247 (48.05%) | 121 (23.54%) |
| Q9-5 T&I related research methods | 9 (1.75%) | 13 (2.53%) | 72 (14.01%) | 288 (56.03%) | 132 (25.68%) |
| Q9-6 T&I industry information | 8 (1.56%) | 4 (0.78%) | 55 (10.70%) | 289 (56.23%) | 158 (30.74%) |
| Q9-7 T&I technology | 8 (1.56%) | 6 (1.17%) | 56 (10.89%) | 285 (55.45%) | 159 (30.93%) |
| Q9-8 T&I project management | 7 (1.36%) | 9 (1.75%) | 85 (16.54%) | 271 (52.72%) | 142 (27.63%) |
| Q9-9 Knowledge of related domains | 10 (1.95%) | 29 (5.64%) | 132 (25.68%) | 224 (43.58%) | 119 (23.15%) |

complement their academic knowledge and understand the specifications of T&I jobs demanded in the market. This echoes the previously-mentioned MTI teachers' needs for T&I practice skills, especially the industry knowledge and T&I technology, which are difficult to obtain in academic institutions and may only be acquired through participation in professional practice.

The items following are related to T&I teaching and research methods, e.g., "T&I related pedagogical methods" (Q9-2) and "T&I related research methods" (Q9-5). It implies that the teachers need to improve their teaching skills, especially how to integrate theories with practice and how to teach efficiently. The items with the lowest score are "theoretical knowledge of T&I studies" (Q9-4), "pedagogic theories" (Q9-1), and "knowledge of related domains" (Q9-9). In comparison to the professional development need of the foreign language teachers of other programs in Chinese universities, i.e., language skills, linguistic theories, and pedagogical methods [4], MTI teachers need more training in professional practice and industry-related training, aligning with the practice-orientation of MTI program.

In terms of training modes, Table 10 shows that the greatest demand of MTI teachers is to participate in "actual T&I practice" (Q10-8). In the qualitative questionnaire, one teacher suggested that we "should be involved in actual T&I practice, project management, and other processes" (20-T38). In addition, as a newly established program, observing experienced teachers' practice and teaching is an effective means of professional development for new MTI teachers, i.e., "to observe and learn from model MTI programs" (Q10-1), "to observe actual T&I practice" (Q10-7) and "to attend lectures of experts" (Q10-2). Some teachers also suggested that "it is more meaningful and rewarding for MTI teachers to learn from more successful MTI programs as a model" (20-T66). In addition, relevant seminars and conferences are also efficient means for MTI teachers to share experiences, such as "to attend teaching seminars" (Q10-4)

**Table 10. MTI teachers' needs of training modes.**

| Items | Strongly Disagree | Disagree | Neutral | Agree | Strongly Agree |
|---|---|---|---|---|---|
| Q10-1 To observe and learn from model MTI programs | 4 (0.78%) | 6 (1.17%) | 60 (11.67%) | 273 (53.11%) | 171 (33.27%) |
| Q10-2 To attend lectures of experts | 6 (1.17%) | 8 (1.56%) | 76 (14.79%) | 280 (54.47%) | 144 (28.02%) |
| Q10-3 To attend online training | 3 (0.58%) | 10 (1.95%) | 119 (23.15%) | 265 (51.56%) | 117 (22.76%) |
| Q10-4 To attend teaching seminars | 5 (0.97%) | 10 (1.95%) | 80 (15.56%) | 283 (55.06%) | 136 (26.46%) |
| Q10-5 To attend academic conferences | 6 (1.17%) | 13 (2.53%) | 83 (16.15%) | 271 (52.72%) | 141 (27.43%) |
| Q10-6 To attend in-house training in T&I firms | 7 (1.36%) | 27 (5.25%) | 95 (18.48%) | 250 (48.64%) | 135 (26.26%) |
| Q10-7 To observe actual T&I practice | 9 (1.75%) | 13 (2.53%) | 57 (11.09%) | 280 (54.47%) | 155 (30.16%) |
| Q10-8 To participate in actual T&I practice | 8 (1.56%) | 8 (1.56%) | 44 (8.56%) | 273 (53.11%) | 181 (35.21%) |

and "to attend academic conferences" (Q10-5). In this category, the relatively lower scores were "to attend online training" (Q10-3) and "to attend in-house training in T&I firms" (Q10-6), and some teachers in the interview suggested that teachers should undertake more in-house training or participate in actual practice in T&I enterprises on a part-time basis, which is also previously suggested by other researchers [38]. However, in practice, due to teachers' time and energy constraints, administrative policies, and other objective circumstances, a systematic administrative system supporting any kind of in-house training in T&I firms has not been developed in MTI teachers' education and sustainable development programs.

## Findings and implications

On the basis of the above analysis and discussion of MTI teachers' perceptions of dilemmas and needs in their professional development, we draw the following findings and implications. Firstly, MTI teachers are basically satisfied with the current status quo of MTI education, but some challenges in teaching, especially those caused by the varying levels of students in MTI education, insufficient number of qualified teachers, limited teaching resources, and the discrepancy between curriculum design and market demand, aroused the dilemma and concern for MTI teachers. While the MTI program is developing rapidly, corresponding measures and policies should be further implemented to formulate more reasonable enrollment requirements, enhance the quality of teaching, and adjust the curriculum to fit the market's needs. As some scholars suggested, the rapid development of the MTI program cannot be at the expense of quality [39].

Secondly, it reveals that the most pressing need of MTI teachers' professional development currently is to increase their knowledge about T&I industry and engagement in T&I practice, which are consistently believed to be the vital guarantee for the quality of MTI education. As previously reiterated, most MTI teachers are faculty members who just shifted from research-oriented postgraduate programs to practice-oriented MTI programs. They are fully aware of the necessity of enriching their T&I practice experience but do not have a clear and specific plan to implement this goal. Moreover, the constraints and entanglement of heavy teaching loads, high expectations of research engagement, and other environmental factors, as well as some personal factors, decrease their motivation to actively participate in T&I practice. Though MTI teachers are aware of the importance of T&I practice, the problems of how to engage in practice and how to resolve the contradiction between their identities as practice-oriented and research-oriented teachers perplexed them. Some scholars have attempted to explore this issue in recent years [22, 34, 40, 41], but there is still no effective solution to the problem. It is suggested that in the future, some practical measures and policies to accommodate the needs of MTI teachers for T&I practice and proper teacher evaluation mechanisms to encourage teachers' engagement in professional practice should be adopted.

Thirdly, regarding in-service training, the study reveals that most of the MTI teachers have had opportunities for in-service training and are generally satisfied with the training, but the needs of MTI teachers in terms of training have not been comprehensively considered. The current T&I teacher training mainly focuses on the teaching philosophy, principles, methods, and curriculum design, emphasizing the theoretical knowledge input and demonstrating successful teaching modes. However, the knowledge transmitted in the existing training does not come from teachers' personal teaching experiences, and simply providing teachers with an imitable or authoritative model to follow is not sufficient [5, 42]. Meanwhile, the training at the technical level neglects the personal development of MTI teachers as "whole persons" [3, 43] and the cultivation of comprehensive literacy, and fails to develop a systematic professional knowledge system for teachers. Therefore, it is recommended that future training should focus on the all-round development of the whole MTI teacher community while at the same time

caring for ongoing professional development needs of the individual teachers. It is necessary to consider not only MTI teachers' initial needs of simply following a model, but also the long-term needs of sustainable professional development.

Moreover, the study finds that most teachers expressed the needs for teamwork support in teaching and research. As a new and rapidly developing program, the faculty is expanding, and many "novice teachers" who are newly shifted to or recruited as MTI teachers urgently need help and support in their professional development. However, while seeking external assistance, MTI teachers should also develop internal motives for their professional development. MTI teachers vary in their educational backgrounds, practice experiences, research fields, and knowledge structures. These differences are valuable resources for teachers to establish a learning community of mutual respect, trust, reciprocity, equality, and openness [44, 45]. Through teachers' interactions and mutual assistance in research, practice, and teaching, they would complement each other in their professional development. For example, teachers with robust research competence, teachers with rich experience in T&I practice, and teachers with rich teaching experience may cooperate with and assist each other as one community of practice. Building a learning community is one of the most efficient approaches to the maintenance of the professional development of tertiary teachers [46, 47].

Fifthly, the environment is one of the most significant factors influencing teachers' professional development [48–50]. The study finds that some environmental factors demotivate MTI teachers from their development, especially the requirement for their research productivity and promotion mechanism. Research engagement is a significant criterion in the evaluation and promotion system of the academic faculty in Chinese universities. However, their engagement in translation/interpreting and teaching practice inhibits their engagement in research. Therefore, the environmental factors should adapt to the characteristics and needs of MTI teachers. For example, to accredit the experience of MTI teachers in T&I practice in the evaluation system, to expand the range of academically recognized journals and papers related to T&I practice, to recognize the T&I projects undertaken by MTI teachers, to establish learning community of practice among MTI teachers, to differentiate promotion criteria for research-oriented and practice-orientated teachers, to balance the requirement of teaching, researching and T&I practice for MTI teachers. An encouraging environment would strengthen their professional identity and enhance their motivation for sustainable professional development [47, 51].

What needs to emphasize is that, compared with the faculty of research-oriented programs, MTI teachers' professional development follows a "trinity" model composed of teaching, academic research, and T&I practice. However, the study indicates that the relationship among the three components is mutually repulsive instead of mutually supportive, which disperses MTI teachers' time and energy, creates contradiction and entanglement, and impedes their healthy, comprehensive and sustainable development. How to improve the "mutual transferability" [24, 52] among research, teaching, and T&I practice and drive the three cogs under the same mechanism progress simultaneously (Fig 1) is key to MTI teachers' sustainable development. The development of an encouraging environment, proper positioning of T&I program, improvement of the quality of MTI curriculum, increase of the opportunities of the professional training, construction of a cooperative learning community, support of promotive evaluation system, and construction of a "whole-person" development concept are all the lubricants that enable the smooth progress of the "three-in-one" mechanism.

## Conclusion

The development of MTI teachers and MTI education are interdependent, and so are the quality of teaching and learning. Hence the exploration of teacher development serves the twin

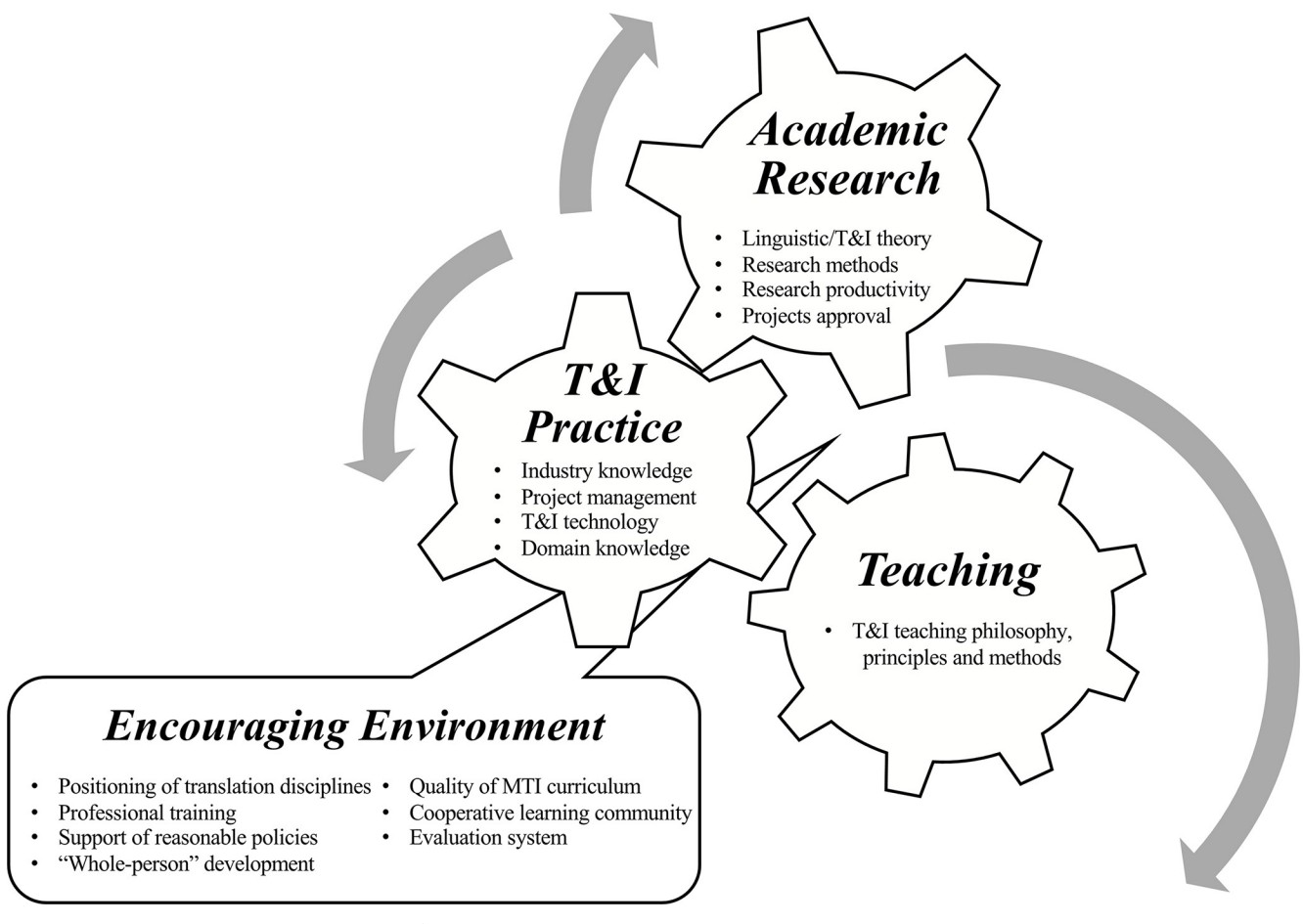

**Fig 1. Three-in-one mechanism of MTI teachers' professional development.**

purposes of enhancing MTI teacher professional learning and improving the quality of MTI education. However, as a newly-established program, the teachers encountered more dilemmas and uncertainties, and the traditional "reference road-map" of foreign language teachers in other academic programs does not fit the needs of their professional development. How to construct a harmonious "three-in-one" model of teaching, research, and T&I practice is currently a significant issue in the field of MTI education. Ultimately, this research will be more meaningful when the results can find more empirical research conducted on MTI teachers in the future, especially from the perspective of constructivism, which focuses on the lived experiences and the inner voice of teachers as an individual person, and can also find more implementations of constructive policies and training to stimulate the enthusiasm and initiative of MTI teachers, enhance the professional competence of MTI teachers, and promote the sustainable development of MTI education. Apart from the core findings and implications, this research has two major limitations. First, we did not further categorize MTI teachers based on their specific pathway (translation or interpreting), making their respective needs and dilemmas of both groups remain underexplored. Moreover, this study adopts qualitative interviews with seven MTI teachers as supplementary to triangulate the quantitative findings. We also recognize the necessity of conducting in-depth qualitative studies in the future on MTI teachers to explore the innermost needs in their professional development.

## Supporting information

**S1 Data.**
(XLSX)

**S1 File.**
(DOCX)

## Author Contributions

**Conceptualization:** Yi Liu.

**Formal analysis:** Yi Liu, Jianyu Liu.

**Funding acquisition:** Yi Liu.

**Investigation:** Yi Liu.

**Methodology:** Yi Liu, Jianyu Liu.

**Software:** Jianyu Liu.

**Visualization:** Jianyu Liu.

**Writing – original draft:** Yi Liu.

**Writing – review & editing:** Yi Liu, Jianyu Liu.

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
