## [Decision Letter · Decision Letter 0]

7 Jun 2023

PONE-D-23-05313Translation and interpreting teachers’ perceptions of dilemma and needs in their professional developmentPLOS ONE

Dear Author(s),

Thank you for submitting your manuscript to PLOS ONE. After careful consideration, we feel that it has merit but does not fully meet PLOS ONE’s publication criteria as it currently stands. Therefore, we invite you to submit a revised version of the manuscript that addresses the points raised during the review process.

Reviewer 1

The “Introduction” part is too lengthy. There should be a separate “Literature Review” Part. The paper should exhibit an adequate understanding of the relevant literature in the field. There is a need to update recent literature up to 2023. The “Methodology” part is well-written and able to meet the objectives of the paper. Conclusions are presented in an appropriate fashion and are supported by the data. The article is presented in an intelligible fashion and is written in standard English.

Reviewer 2

The manuscript appears to be a technically sound piece of scientific research. The conclusions were drawn appropriately based on the data presented. However, inclusion of "Limitation" of this study would add value and is suggested accordingly. Statistical Analyses required to address both the qualitative and quantitative data were presented appropriately. Underlying data were presented in the manuscript as well as in the appendix. It appears that the manuscript was written in a clear, correct and unambiguous way. It is suggested that the author should include "Research Objective" section and the "Research Questions" be moved out from the "Methodology" section and placed ahead of the "Methodology".

Reviewer 1

The “Introduction” part is too lengthy. There should be a separate “Literature Review” Part. The paper should exhibit an adequate understanding of the relevant literature in the field. There is a need to update recent literature up to 2023. The “Methodology” part is well-written and able to meet the objectives of the paper. Conclusions are presented in an appropriate fashion and are supported by the data. The article is presented in an intelligible fashion and is written in standard English.

Reviewer 2

The manuscript appears to be a technically sound piece of scientific research. The conclusions were drawn appropriately based on the data presented. However, inclusion of "Limitation" of this study would add value and is suggested accordingly. Statistical Analyses required to address both the qualitative and quantitative data were presented appropriately. Underlying data were presented in the manuscript as well as in the appendix. It appears that the manuscript was written in a clear, correct and unambiguous way. It is suggested that the author should include "Research Objective" section and the "Research Questions" be moved out from the "Methodology" section and placed ahead of the "Methodology".

We look forward to receiving your revised manuscript.

Kind regards,

Syed Far Abid Hossain, PhD

Academic Editor

PLOS ONE

Journal Requirements:

2. Please provide additional details regarding ethical approval in the body of your manuscript. In the Methods section, please ensure that you have specified the name of the IRB/ethics committee that approved your study."

3. Please provide additional details regarding participant consent. In the Methods section, please ensure that you have specified (1) whether consent was informed and (2) what type you obtained (for instance, written or verbal). If your study included minors, state whether you obtained consent from parents or guardians. If the need for consent was waived by the ethics committee, please include this information.

Reviewers' comments:

Reviewer's Responses to Questions

**Comments to the Author**

1. Is the manuscript technically sound, and do the data support the conclusions?

Reviewer #1: Yes

Reviewer #2: Yes

2. Has the statistical analysis been performed appropriately and rigorously? 

Reviewer #1: Yes

Reviewer #2: Yes

3. Have the authors made all data underlying the findings in their manuscript fully available?

Reviewer #1: Yes

Reviewer #2: Yes

4. Is the manuscript presented in an intelligible fashion and written in standard English?

Reviewer #1: Yes

Reviewer #2: Yes

5. Review Comments to the Author

Reviewer #1: The “Introduction” part is too lengthy. There should be a separate “Literature Review” Part. The paper should exhibit an adequate understanding of the relevant literature in the field. There is a need to update recent literature up to 2023. The “Methodology” part is well-written and able to meet the objectives of the paper. Conclusions are presented in an appropriate fashion and are supported by the data. The article is presented in an intelligible fashion and is written in standard English.

Reviewer #2: The manuscript appears to be a technically sound piece of scientific research. The conclusions were drawn appropriately based on the data presented. However, inclusion of "Limitation" of this study would add value and is suggested accordingly. Statistical Analyses required to address both the qualitative and quantitative data were presented appropriately. Underlying data were presented in the manuscript as well as in the appendix. It appears that the manuscript was written in a clear, correct and unambiguous way. It is suggested that the author should include "Research Objective" section and the "Research Questions" be moved out from the "Methodology" section and placed ahead of the "Methodology".

6. PLOS authors have the option to publish the peer review history of their article (what does this mean?). If published, this will include your full peer review and any attached files.

Reviewer #1: **Yes: **Sayed Farrukh Ahmed

Reviewer #2: No

---

## [Author Response · Author response to Decision Letter 0]

20 Jun 2023

Response to Reviewer 1:

Thank you very much for your constructive suggestions. In the revised manuscript, we reorganized the “Introduction” part (Lines 34-62) and added a separate “Literature Review” part (Lines 63-99). In the “Literature Review” part, we reviewed the literature of the relevant research about translation and interpreting teachers and updated recent literature up to 2023. The changes to these two parts are highlighted in blue in the revised version.

Response to Reviewer 2:

Thank you very much for your positive feedback with constructive suggestions for this manuscript. We have accordingly added the limitations in the “Conclusion” (Lines 467-473). Regarding Comment (2), we placed research questions ahead of the “Methodology” part as suggested. In our revision, according to the structure and logic of the manuscript, we placed research objectives and questions at the end of the “Literature Review” (Lines 93-99). The changes to these parts are also highlighted in blue in the revised version.

---

## [Decision Letter · Decision Letter 1]

21 Dec 2023

PONE-D-23-05313R1Translation and interpreting teachers’ perceptions of dilemma and needs in their professional developmentPLOS ONE

Dear Dr. Liu,

Thank you for submitting your manuscript to PLOS ONE. After careful consideration, we feel that it has merit but does not fully meet PLOS ONE’s publication criteria as it currently stands. Therefore, we invite you to submit a revised version of the manuscript that addresses the points raised during the review process.

Your manuscript has been evaluated by six reviewers: the two previous reviewers (Reviewers 1 and 2) and four new reviewers (Reviewers 3-6); their comments are appended below. Some of the reviewers have commented on opportunities to improve the contextualization for this study, as well as in the reporting of results and statistics, and in clarifying the specific addition this work makes to the literature on this topic. Please ensure you address each of the reviewers' comments when revising your manuscript.

We look forward to receiving your revised manuscript.

Kind regards,

Hugh Cowley

Staff Editor

PLOS ONE

Reviewers' comments:

Reviewer's Responses to Questions

**Comments to the Author**

1. If the authors have adequately addressed your comments raised in a previous round of review and you feel that this manuscript is now acceptable for publication, you may indicate that here to bypass the “Comments to the Author” section, enter your conflict of interest statement in the “Confidential to Editor” section, and submit your "Accept" recommendation.

Reviewer #1: All comments have been addressed

Reviewer #2: All comments have been addressed

Reviewer #3: (No Response)

Reviewer #4: (No Response)

Reviewer #5: All comments have been addressed

Reviewer #6: All comments have been addressed

2. Is the manuscript technically sound, and do the data support the conclusions?

Reviewer #1: Yes

Reviewer #2: (No Response)

Reviewer #3: Yes

Reviewer #4: Partly

Reviewer #5: Partly

Reviewer #6: Yes

3. Has the statistical analysis been performed appropriately and rigorously? 

Reviewer #1: Yes

Reviewer #2: (No Response)

Reviewer #3: Yes

Reviewer #4: No

Reviewer #5: Yes

Reviewer #6: Yes

4. Have the authors made all data underlying the findings in their manuscript fully available?

Reviewer #1: Yes

Reviewer #2: (No Response)

Reviewer #3: Yes

Reviewer #4: Yes

Reviewer #5: Yes

Reviewer #6: Yes

5. Is the manuscript presented in an intelligible fashion and written in standard English?

Reviewer #1: Yes

Reviewer #2: (No Response)

Reviewer #3: Yes

Reviewer #4: Yes

Reviewer #5: Yes

Reviewer #6: Yes

6. Review Comments to the Author

Reviewer #1: All comments have been addressed properly. The revised version is well-written and eligible to publish.

Reviewer #2: (No Response)

Reviewer #3: 1. Apart from research gaps, what are the main facts happening in the field currently? This has not been specifically stated in the introduction.

2. In the discussion section, there is no confirmation of the novelty of the research. This needs to be in the form of a special paragraph.

3. The results of quantitative data processing are only descriptive. Are there no other tests as supporting data for a finding in the quantitative aspect ?

Reviewer #4: The work presents significant improvements, making it easier to understand the study's objectives as well as its limitations, but it still needs further refinement.

Since the analysis of responses is too simplistic, only presenting percentages and averages, it is crucial for the work to be well-grounded in theoretical terms. In this regard, in the literature review, it is not sufficient to merely identify the conducted studies; it is also necessary to present the main conclusions of recent studies within the same scope as this work, and also highlighting the relevance and added value of this study.

In the final part of the introduction, when discussing the studies conducted by Liu & Xu and Liu & Zhang, the authors focus on the methodologies used in these studies rather than their conclusions and/or limitations. The text related to methodology should be placed under the subtopic "Methodology."

Questionnaire: Although reference is made to a previous study, here it is important to specify the type of response (5-point Likert scale: 1- ..., 2- ..., 3-..., 4- ..., and 5- ...).

Sample: It is necessary identify the strata used.

The designation of the figures, e.g., "Fig 2. Descriptive statistics of MTI teachers’ recognition of teaching problems (Mean=3.61)" does not present descriptive statistics (not plural), only one descriptive statistic – the mean. And the mean=3.61 does not make sense to be here, as it is a general average.

Mean: The mean is not the best measure to represent this data. For example, if you calculate the mean of the data presented in the pie charts, you may see that a lot of information is lost. The pie chart allows for a richer and more accurate analysis. Since there are many variables, I understand that you may have opted for bar charts. If making charts is not possible, you can present the values in a table with percentages per question. Suggestion: Instead of bar charts, consider using box-plots.

Bar charts: In the initial ones, the vertical axis displays values 0, 1, 2, 3, 4, and 5. However, the responses vary between 1 and 5. The final charts already range from 1 to 5. The vertical axis (from 1 to 5) does not refer to the mean but rather to the values of the Likert scale. The height of the bars pertains to the mean, but as mentioned earlier, it is not the best measure in this case. A legend is also necessary, as question Q1 differs from chart to chart. It is not enough to be identified in the text; a legend should appear directly on the chart to identify each question.

Is figure 9 your own creation, or is it based on some other work? If the latter is the case, it is necessary to give credit to the original authors.

Reviewer #5: Relatively interesting theme and the proposed paper is original (verified by URKUND/OURIGINAL – anti plagiarism software).

The work presents significant improvements, making it easier to understand the study's objectives as well as its limitations.

There must be a reinforcement of the theoretical framework.

Greater attention should be paid to the identification of figures and data processing. the article must be fully revised.

In the discussion section, there is no confirmation of the novelty of the research. The results of quantitative data processing are only descriptive.

Reviewer #6: The article "Translation and interpreting teachers’ perceptions of dilemma and needs in their professional development" has been well written. Authors have given sufficient background to the study. The methods used are appropriate to the title of the study. The findings provided have addressed the research questions under study.

7. PLOS authors have the option to publish the peer review history of their article (what does this mean?). If published, this will include your full peer review and any attached files.

Reviewer #1: **Yes: **Sayed Farrukh Ahmed

Reviewer #2: No

Reviewer #3: **Yes: **Yandra Rivaldo

Reviewer #4: No

Reviewer #5: No

Reviewer #6: No

---

## [Author Response · Author response to Decision Letter 1]

10 Mar 2024

Yi Liu and Jianyu Liu

Liaoning University

Shenyang, P. R. China

liuyi@lnu.edu.cn, ljy_0923@163.com

9 Jan, 2024

Dear Editors and Reviewers:

Thank you for your letter and for the reviewers’ comments concerning our manuscript entitled “Translation and interpreting teachers’ perceptions of dilemma and needs in their professional development” (ID: PONE-D-23-05313R1). Those comments are all valuable and very helpful for revising and improving our paper, as well as the important guiding significance to our researches. We have studied comments carefully and have made correction which we hope meet with approval. Revised portion are marked in blue in the paper. The main corrections in the paper and the responds to the reviewer’s comments are as flowing:

Responds to the reviewer’s comments:

Reviewer #1: 

All comments have been addressed properly. The revised version is well-written and eligible to publish.

Response to reviewer #1:

Thank you so much for taking the time to review this manuscript.

Reviewer #2: 

(No Response)

Reviewer #3: 

1. Apart from research gaps, what are the main facts happening in the field currently? This has not been specifically stated in the introduction.

2. In the discussion section, there is no confirmation of the novelty of the research. This needs to be in the form of a special paragraph.

3. The results of quantitative data processing are only descriptive. Are there no other tests as supporting data for a finding in the quantitative aspect?

Response to reviewer #3:

Thank you very much for your constructive suggestions. Responses to your suggestions are as following:

1. As stated in the Introduction part, Master of Translation and Interpreting (MTI) as a newly established professional post-graduate program, most MTI teachers were shifted from the academic postgraduate programs of linguistics, applied linguistics, or literature and did not have sufficient translation or interpreting practice and professional expertise essential to the practice-oriented post-graduate program. And this insufficiency brings criticism on the professional development of MTI teachers which are unanimously perceived as the primary guarantee and core driving force for sustainable development of MTI education. Therefore, it is of practical significance to understand the difficulties and needs of MTI teachers in their professional development from their current situation.

2. At the end of the discussion section, based on the results of quantitative data analysis and the supplementation of qualitative interviews, this study proposes a three-in-one model for the professional development of MTI teachers, which is the innovation of this study (see Figure 9 and Lines 462-476).

3. Thank you for your questions and suggestions. Since the objectives of this research is to understand the current situation of difficulties and needs in the professional development of MTI teachers, this study only conducted descriptive analysis on the relevant data, and interpreted the relevant findings in combination with in-depth interviews with 7 teachers. Other quantitative statistical analysis methods (such as t-test, variance analysis, regression analysis, etc.) were not used. However, in the future, we will further explore other issues in this field and enrich the use of statistical methods in research.

Thank you again for your good comments!

Reviewer #4: 

The work presents significant improvements, making it easier to understand the study's objectives as well as its limitations, but it still needs further refinement.

1. Since the analysis of responses is too simplistic, only presenting percentages and averages, it is crucial for the work to be well-grounded in theoretical terms. In this regard, in the literature review, it is not sufficient to merely identify the conducted studies; it is also necessary to present the main conclusions of recent studies within the same scope as this work, and also highlighting the relevance and added value of this study.

2. In the final part of the literature review, when discussing the studies conducted by Liu & Xu and Liu & Zhang, the authors focus on the methodologies used in these studies rather than their conclusions and/or limitations. The text related to methodology should be placed under the subtopic “Methodology.”

3. Questionnaire: Although reference is made to a previous study, here it is important to specify the type of response (5-point Likert scale: 1- ..., 2- ..., 3-..., 4- ..., and 5- ...).

4. The designation of the figures, e.g., "Fig 2. Descriptive statistics of MTI teachers’ recognition of teaching problems (Mean=3.61)" does not present descriptive statistics (not plural), only one descriptive statistic – the mean. And the mean=3.61 does not make sense to be here, as it is a general average.

5. Mean: The mean is not the best measure to represent this data. For example, if you calculate the mean of the data presented in the pie charts, you may see that a lot of information is lost. The pie chart allows for a richer and more accurate analysis. Since there are many variables, I understand that you may have opted for bar charts. If making charts is not possible, you can present the values in a table with percentages per question. Suggestion: Instead of bar charts, consider using box-plots.

6. Bar charts: In the initial ones, the vertical axis displays values 0, 1, 2, 3, 4, and 5. However, the responses vary between 1 and 5. The final charts already range from 1 to 5. The vertical axis (from 1 to 5) does not refer to the mean but rather to the values of the Likert scale. The height of the bars pertains to the mean, but as mentioned earlier, it is not the best measure in this case. A legend is also necessary, as question Q1 differs from chart to chart. It is not enough to be identified in the text; a legend should appear directly on the chart to identify each question.

7. Is figure 9 your own creation, or is it based on some other work? If the latter is the case, it is necessary to give credit to the original authors.

Response to reviewer #4:

Thank you for your valuable feedback and suggestions on this research. We have carefully considered and made modifications based on your recommendations one by one. Responses to your suggestions are as following:

1. We are very sorry for our negligence of the main conclusions of recent studies. In this revised version, we have added main conclusions of recent relevant studies as well as their limitations. We have further clarified the logic of the literature review section and proposed the research significance and concerns of this study.

2. According to the Reviewer’s suggestion, we have also added key findings and limitations of these two pieces of literature to this revised manuscript.

3. Considering the Reviewer’s suggestion, we have added an introduction to the scale (i.e. 5-point Likert scale) in the research instrument part (see Lines 126-129).

Suggestions 4-6:

It is really true as Reviewer suggested that the mean and the bar charts are not the best measure to represent data of this research. In the current revision, we have adjusted all the bar charts and pie charts in the manuscript to tables with labeled items, in order to more accurately present the analytical results. Please refer to Tables 3-10 for the updated representations.

7. Figure 9 “Three-in-one mechanism of MTI teachers’ professional development” (Line 476) is our own creation based on the major findings of this research.

Special thanks to you for your constructive comments and suggestions!

Reviewer #5:

Relatively interesting theme and the proposed paper is original (verified by URKUND/OURIGINAL – anti plagiarism software). The work presents significant improvements, making it easier to understand the study's objectives as well as its limitations.

1. Greater attention should be paid to the identification of figures and data processing. the article must be fully revised.

2. The results of quantitative data processing are only descriptive.

Response to reviewer #5:

Thank you for your valuable suggestions on this research. Responses to your suggestions are as following:

1. Thank you once again for your constructive feedback on the data analysis and presentation of our research. We have thoroughly reanalyzed and re-presented all the data in the manuscript, and corrected any misuse of “the mean” in the original draft (please refer to Tables 3-10 for details). Your insights have been invaluable in improving the accuracy and clarity of our research.

2. Yes, the results of quantitative data processing are only descriptive. The aim of this research is to investigate the status quo of dilemma and inner-world needs in the professional development of MTI teachers. In this study, a descriptive analysis was solely conducted on the pertinent data, with interpretations of the intensive interviews with seven MTI teachers as supplement. Other quantitative statistical analysis techniques, such as t-tests, variance analysis, regression analysis, etc., were not employed. Nevertheless, we plan to delve deeper into related topics in the future and enhance the utilization of statistical methods in our research.

Thank you again for your constructive suggestions!

Reviewer #6: 

The article “Translation and interpreting teachers’ perceptions of dilemma and needs in their professional development” has been well written. Authors have given sufficient background to the study. The methods used are appropriate to the title of the study. The findings provided have addressed the research questions under study.

Response to reviewer #6:

Thank you so much for taking the time to review this manuscript.

We tried our best to improve the manuscript and made some changes in the manuscript. These changes will not influence the content and framework of the paper. 

We appreciate for Editors/Reviewers’ warm work earnestly, and hope that the correction will meet with approval.

Once again, thank you very much for your comments and suggestions.

Both authors have read and approved the revised manuscript. We hope that our resubmission is now suitable for publication in PLOS ONE and we look forward to hearing from you. 

Most sincerely yours, 

Yi Liu and Jianyu Liu

---

## [Decision Letter · Decision Letter 2]

12 Jun 2024

PONE-D-23-05313R2Translation and interpreting teachers’ perceptions of dilemma and needs in their professional developmentPLOS ONE

Dear Dr. Liu,

Thank you for submitting your manuscript to PLOS ONE. After careful consideration, we feel that it has merit but does not fully meet PLOS ONE’s publication criteria as it currently stands. Therefore, we invite you to submit a revised version of the manuscript that addresses the points raised during the review process.

Your manuscript has been re-evaluated by four of the previous reviewers, all of whom are satisfied that their concerns have been addressed. However, I have identified a few minor issues that need correction: 1) I was very confused by the results until I realised that the Table headings are incorrect. When you administered your questionnaire, participants responded 1 for "strongly disagree" and 5 for "strongly agree", but the table headings show the descriptors in reverse order (1 for "strongly agree" and 5 for "strongly disagree"). Please could you relabel all the relevant tables? 2) Please provide a copy of the interview guide used for the qualitative component of the study (as a "supporting information" file). 3) Please provide additional details regarding ethical approval in the body of your manuscript. In the Methods section, please ensure that you have specified the name of the IRB/ethics committee that approved your study.

4) Please provide additional details regarding participant consent. In the Methods section, please ensure that you have specified (1) whether consent was informed and (2) what type you obtained (for instance, written or verbal). If the need for consent was waived by the ethics committee, please include this information.

5) Please report in the Methods section the day, month and year of the start and end of the recruitment period for this study.

We look forward to receiving your revised manuscript.

Kind regards,

Steve Zimmerman, PhD

Senior Editor, PLOS ONE

Journal Requirements:

Reviewers' comments:

Reviewer's Responses to Questions

**Comments to the Author**

1. If the authors have adequately addressed your comments raised in a previous round of review and you feel that this manuscript is now acceptable for publication, you may indicate that here to bypass the “Comments to the Author” section, enter your conflict of interest statement in the “Confidential to Editor” section, and submit your "Accept" recommendation.

Reviewer #1: All comments have been addressed

Reviewer #2: (No Response)

Reviewer #3: All comments have been addressed

Reviewer #4: All comments have been addressed

2. Is the manuscript technically sound, and do the data support the conclusions?

Reviewer #1: Yes

Reviewer #2: (No Response)

Reviewer #3: Yes

Reviewer #4: (No Response)

3. Has the statistical analysis been performed appropriately and rigorously? 

Reviewer #1: Yes

Reviewer #2: (No Response)

Reviewer #3: Yes

Reviewer #4: (No Response)

4. Have the authors made all data underlying the findings in their manuscript fully available?

Reviewer #1: Yes

Reviewer #2: (No Response)

Reviewer #3: Yes

Reviewer #4: (No Response)

5. Is the manuscript presented in an intelligible fashion and written in standard English?

Reviewer #1: Yes

Reviewer #2: (No Response)

Reviewer #3: Yes

Reviewer #4: (No Response)

6. Review Comments to the Author

Reviewer #1: The revised version is well-written. All suggestions have been addressed properly. Recommended for publication.

Reviewer #2: (No Response)

Reviewer #3: Dear Author

Thank you for your efforts in maximizing the quality of your paper. We hope that the spirit of research and publication will always be your priority for sustainable scientific interests.

Reviewer #4: (No Response)

7. PLOS authors have the option to publish the peer review history of their article (what does this mean?). If published, this will include your full peer review and any attached files.

Reviewer #1: **Yes: **Sayed Farrukh Ahmed

Reviewer #2: No

Reviewer #3: **Yes: **Yandra Rivaldo

Reviewer #4: No

---

## [Author Response · Author response to Decision Letter 2]

19 Jun 2024

Yi Liu and Jianyu Liu

Liaoning University

Shenyang, P. R. China

liuyi@lnu.edu.cn, ljy_0923@163.com

19 June, 2024

Dear Dr. Zimmerman,

Thank you very much for your constructive suggestions for revisions to our manuscript entitled “Translation and interpreting teachers’ perceptions of dilemma and needs in their professional development” (ID: PONE-D-23-05313R2). Those suggestions are all valuable and very helpful for revising and improving our paper, as well as providing important guidance for our research. We have studied all the suggestions carefully and have made correction which we hope meet with approval. The revised portions are marked in blue in the paper. The main corrections in the paper are as follows:

1) The table headings show the descriptors in reverse order.

Response to this issue: 

We are very sorry for our negligence of the table headings and we are very grateful for you to raise this issue. In this revised version, we have relabeled all the table headings in our manuscript, with 1 for “strongly disagree” and 5 for “strongly agree”.

2) please provide a copy of the interview guide used for the qualitative component of the study.

Response to this issue: 

We have submitted the interview guide as a supporting information file to the PLOS ONE’s revision system.

3) Please provide additional details regarding ethical approval (specified the name of the ethics committee), participant consent, and the date of the start and end of the recruitment period of this study in the body of your manuscript. 

Response to this issue: 

In this revision, we supplemented the related information of ethics committee (Line 140), participant consent (Lines 119-122 and Lines 158-161), and the date of the start and end of the recruitment period of this study (Line 139) in the “Methodology” section of the manuscript.

Changes to financial disclosure

Yi Liu received funding from Postgraduate Education Reform Project of Liaoning Province 2022 (Grant Number: LNYJG2022009, URL: https://jyt.ln.gov.cn/). 

We sincerely appreciate all the editors and reviewers’ hard work, and hope that the corrections will meet with approval.

Once again, thank you very much for your comments and suggestions.

Both authors have read and approved the revised manuscript. We hope that our resubmission is now suitable for publication in PLOS ONE and we look forward to hearing from you. 

Most sincerely yours, 

Yi Liu and Jianyu Liu

---

## [Editor Report · Decision Letter 3]

10 Jul 2024

Translation and interpreting teachers’ perceptions of dilemma and needs in their professional development

PONE-D-23-05313R3

Dear Dr. Liu,

We’re pleased to inform you that your manuscript has been judged scientifically suitable for publication and will be formally accepted for publication once it meets all outstanding technical requirements.

Kind regards,

Steve Zimmerman, PhD

Senior Editor, PLOS ONE
---

## [Editor Report · Acceptance letter]

1 Aug 2024

PONE-D-23-05313R3 

PLOS ONE

Dear Dr. Liu, 

I'm pleased to inform you that your manuscript has been deemed suitable for publication in PLOS ONE. Congratulations! Your manuscript is now being handed over to our production team.

Kind regards, 

on behalf of

Dr Steve Zimmerman 

Staff Editor

PLOS ONE